# Improved Biological Responses of Titanium Coating Using Laser-Aided Direct Metal Fabrication on SUS316L Stainless Steel

**DOI:** 10.3390/ma14143947

**Published:** 2021-07-14

**Authors:** Tae-In Kim, Se-Won Lee, Woo-Lam Jo, Yong-Sik Kim, Seung-Chan Kim, Soon-Yong Kwon, Young-Wook Lim

**Affiliations:** 1Department of Orthopaedic Surgery, Davos Hospital, Yongin-si, Gyeonggi-do 17063, Korea; 77linus@daum.net; 2Department of Orthopaedic Surgery, College of Medicine, The Catholic University of Korea, Seoul 06591, Korea; ssewon@naver.com (S.-W.L.); jis25@naver.com (W.-L.J.); yongsik@korea.com (Y.-S.K.); kschb129@naver.com (S.-C.K.); sykwon@catholic.ac.kr (S.-Y.K.); 3Department of Orthopaedic Surgery, Yeouido St. Mary’s Hospital, College of Medicine, The Catholic University of Korea, Seoul 07345, Korea; 4Department of Orthopaedic Surgery, Seoul St. Mary’s Hospital, College of Medicine, The Catholic University of Korea, Seoul 06591, Korea; 5Department of Orthopaedic Surgery, Eunpyeong St. Mary’s Hospital, College of Medicine, The Catholic University of Korea, Seoul 03312, Korea

**Keywords:** titanium, direct metal fabrication, metal 3D printing, stainless steel

## Abstract

Direct metal fabrication (DMF) coatings have the advantage of a more uniform porous structure and superior mechanical properties compared to coatings provided by other methods. We applied pure titanium metal powders to SUS316L stainless steel using laser-aided DMF coating technology with 3D printing. The purpose of this study was to determine the efficacy of this surface modification of stainless steel. The capacity of cells to adhere to DMF-coated SUS316L stainless steel was compared with machined SUS316L stainless steel in vitro and in vivo. Morphological in vitro response to human osteoblast cell lines was evaluated using scanning electron microscopy. Separate specimens were inserted into the medulla of distal femurs of rabbits for in vivo study. The distal femurs were harvested after 3 months, and were then subjected to push-out test and histomorphometrical analyses. The DMF group exhibited a distinct surface chemical composition, showing higher peaks of titanium compared to the machined stainless steel. The surface of the DMF group had a more distinct porous structure, which showed more extensive coverage with lamellipodia from osteoblasts than the machined surface. In the in vivo test, the DMF group showed better results than the machined group in the push-out test (3.39 vs. 1.35 MPa, respectively, *p* = 0.001). In the histomorphometric analyses, the mean bone-to-implant contact percentage of the DMF group was about 1.5 times greater than that of the machined group (65.4 ± 7.1% vs. 41.9 ± 5.6%, respectively; *p* < 0.001). The porous titanium coating on SUS316L stainless steel produced using DMF with 3D printing showed better surface characteristics and biomechanical properties than the machined SUS316L.

## 1. Introduction

Metallic biomaterials have several essential properties, such as high corrosion resistance, biocompatibility, osseointegration, durability, and mechanical strength [1,2]. Titanium (Ti) alloy satisfies these requirements and has several advantages over other materials, such as cobalt-based alloys and SUS316L stainless steel, but is disadvantageous in terms of its cost effectiveness and its manufacturing process [3,4,5].

Cost effectiveness may be the most important factor in the absence of differences in clinical results among metals. Moreover, easier manufacturing processes also result in reduced production costs. Stainless steel has advantages over Ti alloy in terms of its lower cost and ease of manufacture [6,7].

Stainless steel shows good biocompatibility. Austenitic 316L SS is well known for its versatility, owing to its outstanding mechanical properties, such as corrosion resistance. Surface coatings are applied when used in biomedical instruments requiring temporary contact [8]. Even with its decent in vitro corrosion resistance, SS has disadvantages such as material transfer between sliding bodies, oxidation, mechanical mixing, strain-induced martensitic transformation, and poor wear resistance [9]. Furthermore, rather than achieving direct bone–implant contact, there is a propensity for a fibrous tissue interface to develop between stainless steel implants and bone, rendering this material inferior to Ti alloy in the field of biomedical applications [3,4,10].

Surface-engineered materials or coating composites are designed to distinctively improve biological properties, as well as tribological, optical, electrical, and chemical properties, among others. Various trials of coatings materials and coating techniques have been carried out with the aim of reducing wear, corrosion and friction. Titanium (Ti) is a representative substance that is used for biomedical device coatings [11,12]. The typical techniques utilized for coatings on 316L stainless steel include sol–gel methods [12,13], filtered arc deposition methods [11,14], physical vapor deposition methods [15,16], magnetron sputtering [17,18], and plasma nitriding methods [19]. Recently, Shin and Kim et al. presented a laser-engineered net shaping coating technology [20]. This coating technology was developed to overcome the limitations of conventional coating methods. They used laser-aided direct metal fabrication (DMF) with 3D printing technology. Its utility as a surface coating technology for artificial joints was evaluated. DMF technology with 3D printing is a newly developed coating method that is cost-effective and maintains mechanical strength. Idealistic surface profiles such as maximum porosity/roughness and suitable pore size can be achieved with this technology. Furthermore, a solid interface between the coating and the substrate can be achieved despite the different properties of the coating and substrate [20].

We studied whether titanium powder coating on stainless steel with DMF technology could improve biological responses compared to machined SUS316L, as reflected by: (1) the chemical composition of the coating surface; (2) the porosity of the coating surface; (3) the cell morphology; (4) the interfacial shear strength measured by push-out test; and (5) bone histomorphometry.

## 2. Materials and Methods

We compared Ti-coated SUS316L stainless steel (SUS 316L (ASTM F138-19), Titanium Industries, Inc., Rockaway, NJ, USA) using DMF (SUS-DMF) with machined SUS316L stainless steel (SUS-machined) in vitro and in vivo. Two types of stainless steel were subjected to assessment of the chemical composition and structure of the surface in vitro. Two types of stainless steel SUS316L discs with diameter of 12 mm and thickness of 10 mm were manufactured (*n* = 20): (1) SUS-machined (*n* = 10) and (2) SUS-DMF types (*n* = 10). Additionally, two types of stainless steel rods (diameter: 2.7 mm; length: 27 mm) were manufactured (*n* = 32): (1) SUS-machined (*n* = 16) and (2) SUS-DMF types (*n* = 16), for in vivo study.

### 2.1. Manufacturing of the SUS-DMF Specimens

Stainless steel surface was irradiated with a high-powered laser to liquefy and coat pure Ti metal powders (Ti powder: 45–150 μm, ASTM F1580-18 (grade 2), Advanced Powders and Coatings, Inc. (AP&C), Boisbriand, QC, Canada). A 3D computer-assisted design (CAD) program was used to construct the porous structure; materials with similar properties of cancellous bone were matched to confer acceptable fixation force [20]. To protect the porous structure from oxidation during the manufacturing process, continuous argon gas was used for the shielding. The gas flow rate for shielding was 7 L/min.

The first coating layer, with an average thickness of 300 μm, was laminated, with the subsequent second layer having an average thickness of 500 μm. The variation in the thicknesses confers further irregularity in the coating profiles.

An optical interferometer (Accura 20001; Interplus corporation, Seoul, Korea) was used to measure the surface roughness. The average surface roughness was 6.1 ± 0.23 μm [mean ± standard deviation] for the SUS-DMF group and 0.2 ± 0.14 μm for the SUS-machined group. After laminating the test samples with platinum, they were analyzed with scanning electron microscopy (SEM; JSM-6700F; JEOL Ltd., Tokyo, Japan).

### 2.2. Surface Chemical Composition and Porosity

After coating the test specimens, the chemical composition and porosity of two surfaces were examined using energy-dispersive X-ray spectroscopy (EDS; xFlash 6i100, Bruker, Billerica, MA, USA) and SEM, respectively.

### 2.3. Culture and Osteogenic Differentiation process of Human Mesenchymal Stem Cells (hMSCs)

Bone marrow-derived hMSCs at passage 2 were acquired from the Catholic Institute of Cell Therapy (Seoul, Korea). Markers such as positive markers (CD73 and 90) and negative markers (CD31, 34, and 45) were used for the confirmation of the certificates of analysis for the hMSC phenotype. hMSCs were cultured in Dulbecco’s modified Eagle’s medium (DMEM) (GE Healthcare Hyclone, Salt Lake City, UT, USA) supplemented with 20% fetal bovine serum (FBS) (GE Hyclone) and 1% penicillin/streptomycin (Gibco-BRL, Grand Island, NY, USA) for five passages. The cells were kept at 37 °C for 24 h in a humidified incubator under an atmosphere of 5% CO_2_. Culture of hMSCs at passage 6 led osteogenic differentiation using a Stem Pro Osteogenesis Differentiation Kit (Gibco-BRL). The osteogenesis differentiation basal medium was supplemented with gentamicin reagent. hMSCs were seeded in 6-well culture plates at a density of 3–9 × 10⁴ cells per cm^2^. The media on the culture plate were replaced every 3–4 days, with the total incubation period being 21 days.

### 2.4. Preparation for Cell Morphology

Osteoblasts derived from hMSCs were seeded at 5 × 10⁴ cells on SUS-DMF and SUS-machined specimens. Six hours after seeding of cells, the media were removed, and the cells were cleansed three times with PBS. After adding 2% glutaraldehyde-PBS solution, the cells were stabilized for 2 h and then washed three times with distilled water (DW). At 30-min intervals, the cells were dehydrated using a 50–100% ethanol series. The cells were left at room temperature for complete evaporation of any remaining ethanol. The specimens were coated with platinum, and two surfaces were then examined by SEM.

### 2.5. Implantation of Coated Rods

The experiments were performed in eight full-grown mature New Zealand white rabbits (>3.2 kg). The rabbits were anesthetized by intramuscular administration of ketamine (35 mg/kg) and Rompun (5 mg/kg). With the animal in the supine position, longitudinal incision on each leg was made from 4 cm above the knee joint to 2 cm below. After the patella was turned over laterally, a hole was made in the medial femoral condyle using a 3.5-mm drill bit, which was filled with a rod coated with SUS-DMF. The contralateral femur was filled with a rod coated with SUS-machined, after going through the same process. After replacing the patella in its normal position, the dissected structures were subsequently repaired with Vicryl 2/0 sutures [21]. Twelve weeks later, the distal femurs were harvested in order to perform push-out tests and histological examinations.

Rabbits were placed separately in cages with a wire bottom in a room with controlled temperature and light. All rabbits were allowed to become familiar with the housing facility for 5–7 days prior to the intervention. They were allowed to have ad libitum access to water and food. Animal care, housing, and interventions complied with the protocol approved by the Institutional Animal Care and Use Committee (IACUC) of The Catholic University of Korea. (CUMC-2016-0006-05).

### 2.6. Interfacial Shear Strength Measurement; Push-Out Test

A push-out test was performed to determine the bonding strength to the bone. Every harvested distal femur was cut at both ends of the rod, and foreign bodies were removed. Each test piece was placed in a testing jig to enable the loading and longitudinal axes of the implant to be aligned accurately. The implants were pushed out from the bone sections with a Universal Testing Machine (DTU-900MH, DaeKyung Tech, Incheon, Korea) at a crosshead speed of 1 mm/min. The measured load was divided by the cortex–implant contact area to calculate the interfacial shear strength [21].

### 2.7. Bone Histomorphometry

After dehydrating the harvested bone tissue with alcohol, it was soaked in Technovit 7200 resin (Morphisto, Frankfurt, Germany). The soaked tissue was immersed in paraffin using a light system (Exakt Technologies Inc., Oklahoma City, Oklahoma). The blocks were sliced into sections 200 μm thick with a solid tissue slicer (Struers, Willich, Germany), and each section was stained with hematoxylin and eosin (H & E; Sigma-Aldrich, St. Louis, MI, USA). Microscopic pictures were procured at an original magnification of ×12.5 (Bx51; Olympus, Tokyo, Japan). The percentage of direct contact between the mineralized bone and the stainless steel surface was determined for each harvested specimen, using an integrative eyepiece with parallel sampling lines at a magnification of ×100.

### 2.8. Statistical Analysis

The mean interfacial shear strength and bone-to-implant contact percentage of the two different surfaces were calculated using Wilcoxon’s signed-rank test. All statistical analyses were performed using SPSS ver. 20.0 (IBM Corporation, Armonk, NY, USA). In the performed analyses, *p* < 0.05 was taken as the level of statistical significance.

## 3. Results

The chemical composition of the SUS316L surface in the two groups was evaluated with EDS. Compared to the SUS-machined specimens (Figure 1A), the SUS-DMF type presented a totally contrasting surface with high peaks of Ti (Figure 1B). In the SEM images, the surface of the machined SUS316L possessed a flat, monotonous appearance (Figure 2A). On the other hand, the surface of DMF had a more distinct porous structure than the machined surface (average pore size in the coating layer: 200–500 μm; average porosity: 62.4 ± 7.1%; coating thickness: 500 ± 100 μm) (Figure 2B).

With regard to cell morphology, after a 6-h incubation, the SUS-DMF surfaces were covered with more prominent lamellipodia from the osteoblasts (Figure 3A) than the machined surfaces (Figure 3A). In addition, thin cytoplasmic projections (filopodia) extended into the interior of the pores (Figure 3B).

In the biomechanical push-out test, the ultimate shear strength in the DMF group (3.39 MPa) was 2.5 times greater than that in the machined group (1.35 MPa) (*p* = 0.001) (Figure 4).

The mean bone-to-implant contact percentage in the DMF groups (65.4 ± 7.1%) (Figure 5A) was 1.5 times that of the machined group (41.9 ± 5.6%) (Figure 5B) (*p* < 0.001) in the histomorphometric analysis. The SUS-DMF group showed tighter attachment than the machined group.

## 4. Discussion

The results of the present study demonstrate that Ti-coated stainless steel using DMF with 3D printing produces favorable biological surface characteristics and better biomechanical strength than machined SUS316L in vitro and in vivo.

Selective laser melting and electron beam melting are 3D printing methods that use an energy source to melt and fuse selected regions of the coating powder in accordance with CAD data [22]. The DMF method differs from other methods in how the coating powders are sprayed and laminated onto the surface. The DMF method has advantages over other 3D printing methods in terms of manufacturing time and ease of fabrication [20].

Use of DMF with 3D printing has the advantage of allowing control over the coating surface porosity, which is not possible with conventional methods [20,23]. We have previously reported an average porosity of 65 ± 5% for DMF specimens with thicknesses ranging from 200 to 500 μm, which is a value that is similar to those found for human cancellous bone (50–90%) [20]. Moreover, Ti is highly cytocompatible, which creates a more favorable structure for the attachment and proliferation of osteoblasts [2,24]. Keller et al. reported that surface porosity positively affects both osteoblast attachment and number [25]. Zhu and colleagues also reported that porous structures enhance cell attachment [26]. Lamellipodia and filopodia are two important cytoplasmic protrusions that are closely related to cell migration [27,28]. Migrating cells move using their lamellipodia [29]. Cells on micron- and submicron-scale structures have been observed to enter pores and attach to the substrate via their filopodia [26]. Lee et al. reported that filopodia contribute to cell motility by functioning as antennae for detecting the surroundings of the cell [30]. Consistent with the results of previous studies, our results showed that cells on the surface of the DMF group possessed more prominent lamellipodia and filopodia.

Shear strength also increases, with greater integration between implants and bone, with increased surface porosity [31,32]. Our study indicates that the bonding strength was 2.5 times higher in the highly porous SUS-DMF group than in the SUS-machined group, which was probably related to the higher degree of bone-to-implant contact in the SUS-DMF group. Svehla et al. reported that porous Ti implants possessed greater shear strength than Ti grit-blasted implants or hydroxyapatite-coated Ti grit-blasted implants [32]. Surfaces with greater porosity and roughness enhance the biomechanical characteristics at the interface in bone-anchored implants by providing mechanical interlocking [33]. In our previous study, the DMF group possessed a tensile strength 17.5% greater and a shear strength 10.2% greater than when coatings were applied using Ti plasma spray [20].

There are several limitations to this study. First, we did not compare the DMF method with other 3D printing methods (such as SLM and EBM), and only performed a comparison with machined SUS, so it is uncertain on the basis of the results of this study whether the favorable result of SUS-DMF is because the SUS is coated with a non-SUS material, or is due to the Ti coating. Further studies are required to determine whether this method shows better osseointegration and mechanical strength than other 3D methods. Second, a DNA study was not carried out in this study. Further DNA studies could reinforce these findings by performing an evaluation of the levels of type I collagen and osteocalcin.

## 5. Conclusions

We found that the Ti coating on SUS316L stainless steel manufactured by DMF using the metal 3D printing technique was superior to machined SUS316L with respect to biological responses. The DMF specimens showed better surface characteristics and biomechanical properties than the machined SUS316L, both in vitro and in vivo. In addition, this method is cost-effective and uses an automated manufacturing process. Thus, DMF with metal 3D printing can be applied for many metals for the production and processing of prostheses.

## Figures and Tables

**Figure 1 materials-14-03947-f001:**
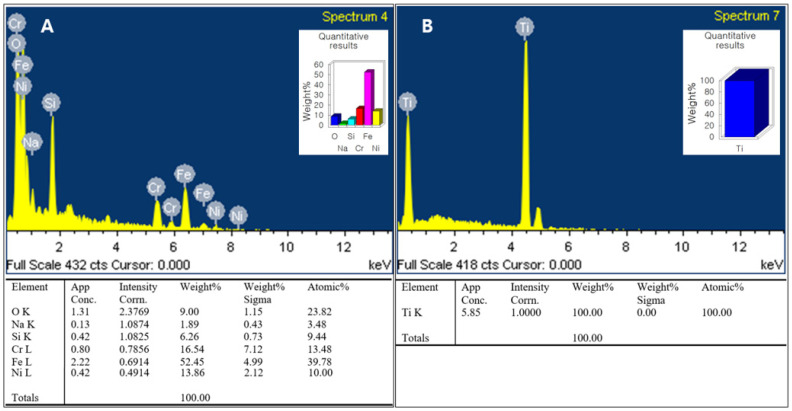
EDS of (**A**) the SUS-machined group and (**B**) the SUS-DMF group. Compared to (**A**) the machined specimen, (**B**) the SUS–DMF group showed an absolutely different pattern, showing high peaks of Ti.

**Figure 2 materials-14-03947-f002:**
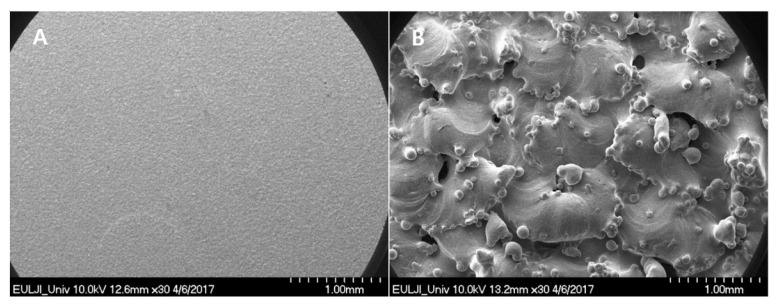
SEM images of the porous surfaces of (**A**) the SUS-machined specimens (×30) and (**B**) the SUS-DMF specimens (×30), showing the different surface properties. Compared with the machined surface, the SUS-DMF specimen exhibited a remarkable porosity, ranging from 200 to 500 μm.

**Figure 3 materials-14-03947-f003:**
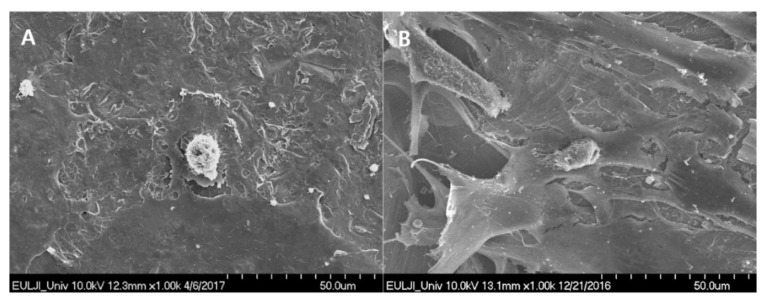
SEM images of osteoblast after 6 h of incubation on (**A**) SUS-machined specimens (×1000) and (**B**) SUS-DMF specimens (×1000). (**A**) The surface of the SUS-machined specimens was covered with sparse, small osteoblast cells. (**B**) The surface of the DMF specimens was firmly covered with large, healthy lamellipodia of osteoblast cells. Branches of thin cytoplasmic process from filopodia were seen entering the pores.

**Figure 4 materials-14-03947-f004:**
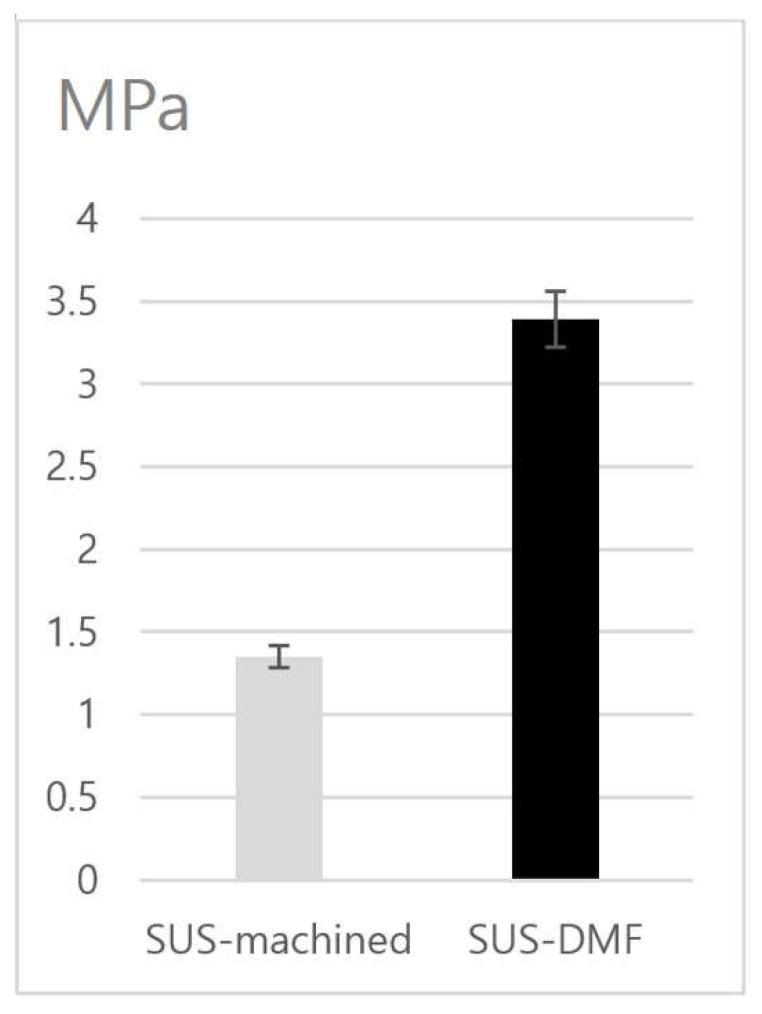
Results of the shear stress of the SUS-machined group and the SUS-DMF group in the push-out test.

**Figure 5 materials-14-03947-f005:**
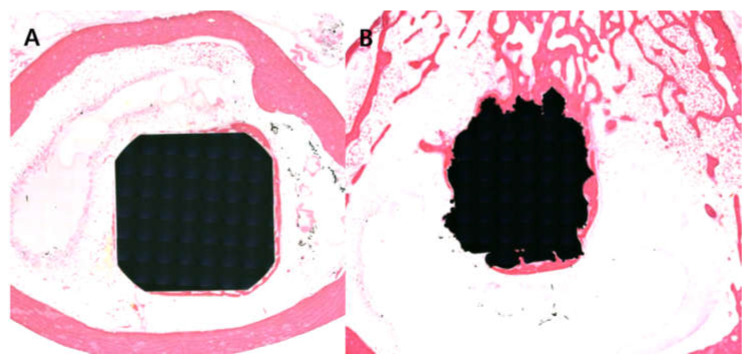
Light microscope images revealing the bone-to-implant contact of (**A**) the SUS-machined (×12.5) and (**B**) SUS-DMF (×12.5) samples. The bone-to-implant contact of the DMF group was superior to that of the machined group.

## Data Availability

The datasets generated during and/or analysed during the current study are available from the corresponding author on reasonable request.

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
