# Peer review of "Improved Biological Responses of Titanium Coating Using Laser-Aided Direct Metal Fabrication on SUS316L Stainless Steel"

_materials, 2021, doi:10.3390/ma14143947_

Round 1
Reviewer 1 Report
The introduction is very limited. In particular, the publications focused on laser machining of Ti and austenitic stainless steels should be given. Now, it is unclear whether similar attempts have been carried out (there has been plenty of them) and whether this scientific problem is justified.
Materials and Methods lack information on the chemical composition of both steels, Ti powder (also on average grain size), and applied laser equipment (name, manufacturer, details of laser deposition. There is neither name nor manufacturer for EDS. The same is true for the push-out tests – give a standard or describe the method. And also for the light microscope. Most importantly, there is no mention of an obligatory allowance for tests on animals (rabbits), please give the number of such a document. If not, such tests are illegal and should not be taken into account.
Results demonstrate also some errors. Figure 4: change Mpa to MPa. Line 211: micrometers, the unit, is improper.
Reviewer 2 Report
Title:
Does Titanium Coating Using Laser-aided Direct Metal Fabrication Improve Biological Responses of SUS316L Stainless Steel?
This manuscript mentioned that the investigation of the biological responses on the Ti-coated SUS316 and SUS316. However, there are some unclear explanations, especially metal preparation and properties. The manuscript needs to be major changed for publication.
- The title is a question form. I wonder that it is inappropriate for an article in the field of Materials because this journal is not a medical journal. This article is not a review one.
- 88, “The average surface roughness (Ra value) was 6.1 µm±0.23 lm”. What the unit of lm? The quantitative values of the surface roughness need to be mentioned to compare the size of the osteoblast cell. The authors need a more detailed discussion about this point.
- Ti coating by laser sintering is difficult because Ti is easily oxidized. In the 3D printing process, what was the atmosphere? What’s the pressure? EDS mapping is shown in Fig.1(B). What’s the concentration of oxygen in the coated Ti? Is it increased after the laser sintering by comparing it with the raw Ti powder? The basic data for the preparation of the samples are unclear.
- The SUS-DMF is effective in push-out test shown in Fig.4. However, the surface of Ti must be oxidized without specific treatment or in the atmosphere. My question is whether the improvement by SUS-DMS was caused by Ti-coating or Ti oxide coating. Furthermore, is the kind of coated materials important or NOT-SUS surface important for the improvement? The authors should mention the mechanism for the improvement.
Round 2
Reviewer 2 Report
Title:
Does Titanium Coating Using Laser-aided Direct Metal Fabrication Improve Biological Responses of SUS316L Stainless Steel?
This manuscript mentioned that the investigation of the biological responses on the Ti-coated SUS316 and SUS316. All the reviewer’s concerns became clear in the revised manuscript. The revised manuscript is sufficient interest for publication.